# The neutrophil-to-lymphocyte ratio as a predictor of all-cause mortality in individuals with anemia: A population-based study

Xiangkuan Cheng[1,2], Lanling Liu[3], Yueming Tian[2], Hong Zhang[1]*, Mingdeng Wang[4]*, Yuansheng Lin[4]*

1 Department of Otolaryngology, Suzhou Research Center of Medical School, Suzhou Hospital, Affiliated Hospital of Medical School, Nanjing University, Suzhou, China, 2 Department Care Unit, Hebei Yanda Lu Daopei Hospital, Langfang, Hebei, China, 3 Department of Respiratory and Critical Care Medicine, Beijing Anzhen Hospital, Capital Medical University, Beijing, China, 4 Department of Intensive Care Unit, Suzhou Research Center of Medical School, Suzhou Hospital, Affiliated Hospital of Medical School, Nanjing University, Suzhou, China

* 2448076169@qq.com (HZ); 1376590818@qq.com (MW); linys202012@163.com (YL)

## Abstract

### Background

This study investigates the association between the neutrophil-to-lymphocyte ratio (NLR) and all-cause mortality in individuals with anemia using the National Health and Nutrition Examination Survey (NHANES) 1999–2018 dataset.

### Methods

We included 3,212 participants with anemia, categorized into three groups by NLR values. Baseline characteristics, comorbidities, and demographics were analyzed. We performed univariate logistic regression, multivariable Cox regression, non-linear regression, and breakpoint analysis to examine NLR-mortality relationships. Sub-group analysis assessed effect modification by clinical factors.

### Results

The mean age of the cohort was 56.0 ± 18.3 years. Participants in the highest NLR tertile (T3) had significantly higher mortality risk, with an HR of 1.25 (95% CI:1.07–1.48, p = 0.007) in the fully adjusted model. Univariate logistic regression showed that NLR was independently associated with mortality (odds ratio [OR] = 1.44, 95% confidence interval [CI]: 1.35–1.52, p < 0.001), with higher mortality risk in the highest NLR tertile (OR = 2.9, 95% CI: 2.39–3.53, p < 0.001). Multivariable Cox regression analysis confirmed NLR as a significant predictor (hazard ratio [HR] = 1.11, 95% CI: 1.07–1.15, p < 0.001). A non-linear regression analysis identified a data-derived threshold at NLR = 1.475, with the risk of mortality increasing significantly above this threshold (HR = 1.134, 95% CI: 1.073–1.2, p < 0.001).

**Data availability statement:** The datasets analyzed in this study are publicly available from the NHANES repository: NHANES 1999–2018 Database: https://wwwn.cdc.gov/nchs/nhanes/. All data are de-identified and accessible without restriction.

**Funding:** This study was supported by Suzhou Science and Education Strengthening Health Project (QNXM2024092) awarded to Yuansheng Lin. The funder had no role in study design, data collection and analysis, decision to publish, or preparation of the manuscript.

**Competing interests:** The authors declare no competing interests.

## Conclusion

NLR is a significant predictor of all-cause mortality in individuals with anemia. Elevated NLR, particularly above 1.475, is associated with a 25.0% higher mortality risk, suggesting its potential utility as a prognostic biomarker in this context.

## Introduction

Anemia is a prevalent condition, particularly among older adults and individuals with chronic diseases, and is associated with an increased risk of adverse health outcomes, including mortality [1–3]. Despite its significance, the role of biomarkers in predicting mortality within the anemia population remains underexplored. One such biomarker is the neutrophil-to-lymphocyte ratio (NLR), a leukocyte-derived inflammation index calculated from routine complete blood count differentials that has gained attention as a potential prognostic marker in various conditions, including cardiovascular diseases [4], cancer [5], and chronic kidney disease (CKD) [6]. Elevated NLR has been associated with poorer outcomes, suggesting its potential to serve as a predictor of mortality in these populations.

However, while NLR has been linked to mortality risk in individuals with various chronic conditions, its association with all-cause mortality in individuals with anemia remains unclear. Anemia, often a manifestation of underlying chronic diseases, may exacerbate the inflammatory response, potentially influencing the prognostic value of NLR [7]. Understanding whether elevated NLR can predict mortality specifically in anemic individuals could provide important clinical insights for managing these patients, particularly those with comorbidities that exacerbate the risk of poor outcomes.

We hypothesized that elevated NLR would independently predict increased all-cause mortality in anemic individuals after adjusting for key comorbidities. This study aims to investigate the association between NLR and all-cause mortality in individuals with anemia using data from the National Health and Nutrition Examination Survey (NHANES) 1999–2018. By analyzing this large, representative dataset, we seek to evaluate whether elevated NLR can serve as a significant independent predictor of mortality in this population, and to explore the potential underlying factors that may modify this relationship.

## Methods

### Study design and population

This retrospective cohort study was conducted and reported in accordance with the STROBE (Strengthening the Reporting of Observational Studies in Epidemiology) guidelines for observational research. This retrospective cohort study utilized data from the NHANES 1999−2018 [8], which is a comprehensive survey conducted by the Centers for Disease Control and Prevention (CDC) to assess the health and nutritional status of non-institutionalized individuals in the United States. The NHANES employs a stratified, multistage probability sampling design to ensure

a nationally representative sample. The study participants were aged ≥20 years, initially including 55,081 individuals. After excluding 1,491 pregnant participants, the study cohort was reduced to 53,590. Participants with missing data on any study variables (hemoglobin, NLR components, covariates, or mortality status) were excluded via complete-case analysis. This approach aligns with STROBE item 12c for handling missing data. It is noteworthy that multiple imputation was not performed for missing data; however, sensitivity analyses conducted on the complete-case dataset confirmed the consistency of the primary findings. Further exclusions for missing data resulted in 37,925 participants eligible for analysis, from which 3,212 participants with anemia were selected for the final study cohort. Participants without anemia (n = 34,713) were excluded. Ethical approval (Protocol #98−12) was secured via the NHANES Institutional Review Board. This retrospective analysis utilized fully anonymized, pre-existing NHANES public-use data. Consistent with U.S. Department of Health and Human Services regulations (45 CFR §46.104(d) [4]) and STROBE item 7, this secondary analysis of de-identified data was exempt from requiring additional IRB review or individual participant consent.

## Sample size & power

Based on the 3,212 anemic adults included in our cohort and the observed all-cause mortality rate. Assuming a two-sided $\alpha = 0.05$ and using the Cox proportional hazards model framework, with a baseline mortality rate of approximately 27.4%, our available sample size provides 90% power to detect a hazard ratio (HR) ≥ 1.25 for the primary outcome.

## Inclusion and exclusion criteria

Anemia was defined according to the World Health Organization (WHO) criteria as hemoglobin levels <13 g/dL for men and <12 g/dL for women [9]. Pregnant women were excluded from the analysis, and participants with missing data on key variables were also excluded. The final study population consisted of 3,212 participants with anemia.

## Baseline characteristics

Baseline characteristics, including demographic factors such as age, gender, ethnicity, marital status, education level, and lifestyle factors such as body mass index (BMI), smoking, and alcohol use, were collected through self-reported questionnaires and interviews. Medical history and comorbid conditions, such as chronic kidney disease (CKD), diabetes mellitus (DM), hypertension, stroke, coronary heart disease (CHD), and cancer, were determined through clinical assessments and participant reports.

## Neutrophil-to-lymphocyte ratio (NLR)

NLR was calculated by dividing the absolute neutrophil count by the absolute lymphocyte count, both of which were obtained from blood samples collected during the NHANES examination [10]. Participants were stratified into three groups based on NLR tertiles: Group 1 (lowest NLR: < 1.60), Group 2 (middle NLR: 1.60–2.46), and Group 3 (highest NLR: ≥ 2.46).

## Mortality data

The primary outcome of this study was all-cause mortality. Mortality data were obtained from the NHANES linked mortality files, which provide information on survival status up until 2018. Mortality data were obtained from the NHANES-linked National Death Index (NDI) files through December 31, 2019, using probabilistic matching with 12 identifiers (including SSN, name, and DOB). This linkage methodology aligns with the RECORD (REporting of studies Conducted using Observational Routinely collected Data) guideline Item 7.1 for validating person-level data linkage. NHANES-NDI linkage validity has been previously established (PMID: 25052282; match sensitivity >98.5%).

## Non-linear and breakpoint analysis

Non-linear relationships between NLR and mortality were evaluated using restricted cubic splines (RCS) with 3 knots (10th, 50th, 90th percentiles). A threshold effect (i.e., a 'breakpoint') was identified via piecewise linear regression, which detects NLR values where the mortality risk slope changes significantly. The term 'breakpoint' refers to the NLR value (1.475) above which each unit increase in NLR confers a 13.4% higher mortality risk (HR = 1.134, p < 0.001), whereas below this threshold, NLR changes showed no significant association (HR = 0.592, p = 0.06).

## Subgroup analysis

Subgroup analysis was performed to explore potential effect modification by factors including gender, age, smoking, drinking, diabetes, and CHD. Multiplicative interaction terms (e.g., NLR × diabetes) were incorporated into the multivariable Cox regression models. HRs with 95% CIs were calculated for each subgroup, and interaction p-values were derived from likelihood ratio tests comparing models with and without interaction terms.

## Statistical analysis

Descriptive statistics were used to summarize the baseline characteristics of the study population. Continuous variables were reported as mean ± standard deviation (SD), and categorical variables were presented as frequencies and percentages. Group comparisons were performed using analysis of variance (ANOVA) for continuous variables and chi-square tests for categorical variables.

Univariate logistic regression analysis was conducted to identify demographic, clinical, and inflammatory factors associated with mortality. Odds ratios (ORs) with 95% confidence intervals (CIs) were calculated for each variable.

Multivariable Cox proportional hazards regression models were used to evaluate the relationship between NLR and all-cause mortality while adjusting for potential confounders. The analysis included a crude model, a model adjusted for age and sex (Model 1), and a model further adjusted for comorbidities such as CKD, DM, hypertension, and stroke (Model 2). The fully adjusted model (Model 3) included all variables of interest.

To examine the non-linear association between NLR and mortality, a non-linear regression analysis was performed. Breakpoint analysis was conducted to identify the critical NLR threshold beyond which the mortality risk significantly increased. A p-value <0.05 was considered statistically significant for all analyses.

# Results

## Baseline characteristics of participants with anemia

The study initially included 55,081 participants aged ≥20 years from the NHANES 1999–2018 dataset (Fig 1). After excluding 1,491 pregnant participants, the study population was reduced to 53,590. Further exclusions for missing data resulted in 37,925 participants eligible for analysis. Among them, 3,212 participants with anemia formed the final study cohort, while 34,713 participants without anemia were excluded.

A total of 3,212 participants with anemia were included in the study (Table 1). The mean age of the cohort was 56.0 ± 18.3 years, with significant age differences across the groups (p < 0.001). Group 1 (n = 1,069) had the youngest mean age (51.8 ± 17.9), while Group 3 (n = 1,071) had the oldest (61.8 ± 17.6). Significant differences were also observed in gender, ethnicity, and the prevalence of comorbidities such as CKD, DM, hypertension, stroke, CHD, and cancer (all p < 0.001). The all-cause mortality rate was highest in Group 3 (40.6%), followed by Group 2 (22.4%) and Group 1 (19.1%) (p < 0.001). Consistent with the comorbidity burden, Group 3 exhibited significantly lower proportions of health-protective behaviors: never-smokers (54.6% vs 65.1% in Group 1, p < 0.001) and never-drinkers (16.4% vs 23.1% in Group 1, p < 0.001). This behavioral gradient parallels the NLR-associated mortality risk pattern.

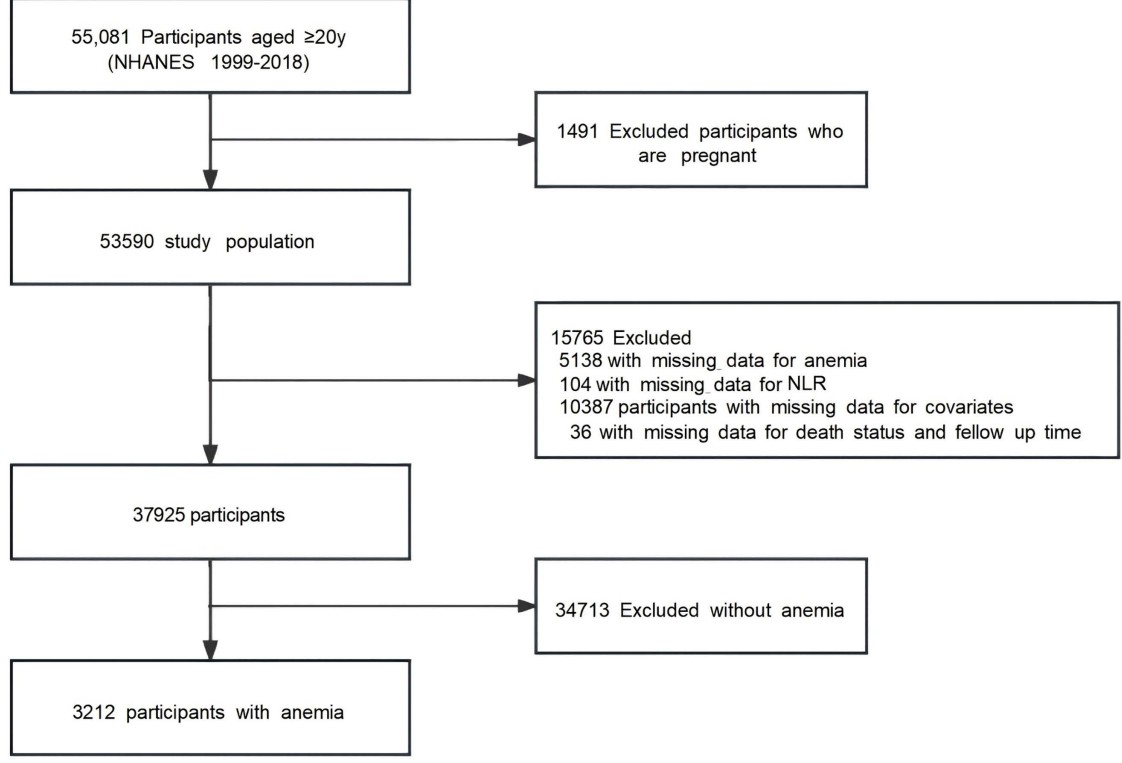

**Fig 1. Flowchart of patient selection.**

## Univariate logistic regression analysis

Univariate logistic regression showed that age, sex, ethnicity, marital status, education level, BMI, smoking, alcohol use, and various comorbidities, including CKD, DM, hypertension, stroke, and CHD, were significantly associated with mortality (all $p < 0.001$) (Table 2). Notably, NLR was independently associated with increased mortality risk (OR = 1.44, 95% CI: 1.35–1.52, $p < 0.001$), particularly in those with higher NLR values, especially in the highest tertile (OR = 2.9, 95% CI: 2.39–3.53, $p < 0.001$). Logistic regression model performance was assessed using the area under the receiver operating characteristic curve (AUC) and the Hosmer-Lemeshow goodness-of-fit test. The univariate model achieved an AUC of 0.673 (95% CI: 0.614–0.659), indicating gooddiscrimination. These values have been added to the Results section for completeness (S1 Fig).

## Multivariable cox regression analysis

In multivariable Cox regression, NLR remained a significant predictor of all-cause mortality after adjusting for confounders (Table 3). In the crude model, the HR for NLR was 1.21 (95% CI: 1.18–1.23, $p < 0.001$). After adjusting for age and sex (Model 1), the HR was 1.12 (95% CI: 1.08–1.16, $p < 0.001$). Further adjustment for comorbidities such as CKD, DM, and hypertension (Model 2) yielded an HR of 1.11 (95% CI: 1.07–1.15, $p < 0.001$). In the fully adjusted model (Model 3), the association remained significant with an HR of 1.11 (95% CI: 1.07–1.15, $p < 0.001$). Participants in the highest NLR tertile (T3) had significantly higher mortality risk, with an HR of 1.25 (95% CI: 1.07–1.48, $p = 0.007$) in the fully adjusted model.

**Table 1. Baseline characteristics of participants with anemia.**

| Variables | Total | Neutrophils/Lymphocyte Ratio (NLR) | | | P-value |
|---|---|---|---|---|---|
| | | Group 1 (NLR<1.60) | Group 2 (1.6–2.46) | Group 3 (≥2.46) | |
| Participants, n | 3212 | 1069 | 1072 | 1071 | |
| Age (year), Mean±SD | 56.0±18.3 | 51.8±17.9 | 54.4±18.0 | 61.8±17.6 | < 0.001 |
| Gender, n (%) | | | | | < 0.001 |
| Male | 1182 (36.8) | 315 (29.5) | 352 (32.8) | 515 (48.1) | |
| Female | 2030 (63.2) | 754 (70.5) | 720 (67.2) | 556 (51.9) | |
| Race, n (%) | | | | | < 0.001 |
| Non-Hispanic White | 964 (30.0) | 188 (17.6) | 296 (27.6) | 480 (44.8) | |
| Non-Hispanic Black | 1348 (42.0) | 590 (55.2) | 446 (41.6) | 312 (29.1) | |
| Mexican American | 432 (13.4) | 134 (12.5) | 161 (15) | 137 (12.8) | |
| Other Hispanic | 224 (7.0) | 66 (6.2) | 87 (8.1) | 71 (6.6) | |
| Other Race – Including Multi-Racial | 244 (7.6) | 91 (8.5) | 82 (7.6) | 71 (6.6) | |
| Marital status, n (%) | | | | | 0.043 |
| Married or living with a partner | 1738 (54.1) | 554 (51.8) | 612 (57.1) | 572 (53.4) | |
| Living alone | 1474 (45.9) | 515 (48.2) | 460 (42.9) | 499 (46.6) | |
| Poverty Income Ratio, Median (IQR) | 1.8 (1.0, 3.5) | 1.8 (1.0, 3.5) | 1.8 (1.0, 3.6) | 1.8 (1.1, 3.3) | 0.747 |
| Education, n (%) | | | | | 0.355 |
| <9th grade | 441 (13.7) | 124 (11.6) | 161 (15) | 156 (14.6) | |
| 9–11th grade | 515 (16.0) | 177 (16.6) | 158 (14.7) | 180 (16.8) | |
| High school | 740 (23.0) | 252 (23.6) | 239 (22.3) | 249 (23.2) | |
| College | 914 (28.5) | 310 (29) | 314 (29.3) | 290 (27.1) | |
| Graduate | 602 (18.7) | 206 (19.3) | 200 (18.7) | 196 (18.3) | |
| Body mass index, kg.m², Mean±SD | 29.9±7.9 | 29.7±7.5 | 30.3±7.9 | 29.8±8.1 | 0.149 |
| Smoker, n (%) | | | | | < 0.001 |
| Never | 1941 (60.4) | 696 (65.1) | 660 (61.6) | 585 (54.6) | |
| Former | 862 (26.8) | 229 (21.4) | 281 (26.2) | 352 (32.9) | |
| Now | 409 (12.7) | 144 (13.5) | 131 (12.2) | 134 (12.5) | |
| Alcohol user, n (%) | | | | | < 0.001 |
| Never | 637 (19.8) | 247 (23.1) | 214 (20) | 176 (16.4) | |
| Former | 768 (23.9) | 212 (19.8) | 240 (22.4) | 316 (29.5) | |
| Mild | 1001 (31.2) | 320 (29.9) | 322 (30) | 359 (33.5) | |
| Moderate | 420 (13.1) | 146 (13.7) | 158 (14.7) | 116 (10.8) | |
| Heavy | 386 (12.0) | 144 (13.5) | 138 (12.9) | 104 (9.7) | |
| WBC (10³/μl), Median (IQR) | 6.5 (5.3, 8.1) | 5.8 (4.7, 7.2) | 6.6 (5.5, 8.1) | 7.2 (6.0, 8.8) | < 0.001 |
| Neutrophils (10³/μl), Mean±SD | 4.1±2.3 | 2.9±1.4 | 4.0±1.2 | 5.3±3.0 | < 0.001 |
| Lymphocyte (10³/μl), Median (IQR) | 1.9 (1.5, 2.4) | 2.3 (1.9, 2.9) | 1.9 (1.6, 2.4) | 1.4 (1.1, 1.8) | < 0.001 |
| Chronic kidney disease, n (%) | | | | | < 0.001 |
| No | 2007 (62.5) | 786 (73.5) | 705 (65.8) | 516 (48.2) | |
| Yes | 1205 (37.5) | 283 (26.5) | 367 (34.2) | 555 (51.8) | |
| Diabetes mellitus, n (%) | | | | | < 0.001 |
| No | 2246 (69.9) | 821 (76.8) | 742 (69.2) | 683 (63.8) | |
| Yes | 966 (30.1) | 248 (23.2) | 330 (30.8) | 388 (36.2) | |
| Hypertension, n (%) | | | | | < 0.001 |
| No | 1387 (43.2) | 521 (48.7) | 492 (45.9) | 374 (34.9) | |
| Yes | 1825 (56.8) | 548 (51.3) | 580 (54.1) | 697 (65.1) | |

*(Continued)*

Table 1. (Continued)

| Variables | Total | Neutrophils/Lymphocyte Ratio (NLR) | | | P-value |
| --- | --- | --- | --- | --- | --- |
| | | Group 1 (NLR < 1.60) | Group 2 (1.6–2.46) | Group 3 (≥2.46) | |
| Stroke, n (%) | | | | | < 0.001 |
| No | 2975 (92.6) | 1015 (94.9) | 1012 (94.4) | 948 (88.5) | |
| Yes | 237 (7.4) | 54 (5.1) | 60 (5.6) | 123 (11.5) | |
| Coronary heart disease, n (%) | | | | | < 0.001 |
| No | 2958 (92.1) | 1017 (95.1) | 1011 (94.3) | 930 (86.8) | |
| Yes | 254 (7.9) | 52 (4.9) | 61 (5.7) | 141 (13.2) | |
| Cancer, n (%) | | | | | < 0.001 |
| No | 2796 (87.0) | 971 (90.8) | 947 (88.3) | 878 (82) | |
| Yes | 416 (13.0) | 98 (9.2) | 125 (11.7) | 193 (18) | |
| Thyroid disease, n (%) | | | | | 0.207 |
| No | 2814 (87.6) | 947 (88.6) | 944 (88.1) | 923 (86.2) | |
| Yes | 398 (12.4) | 122 (11.4) | 128 (11.9) | 148 (13.8) | |
| All-Cause mortality, n (%) | | | | | < 0.001 |
| Alive | 2333 (72.6) | 865 (80.9) | 832 (77.6) | 636 (59.4) | |
| Death | 879 (27.4) | 204 (19.1) | 240 (22.4) | 435 (40.6) | |

## Non-linear and breakpoint analysis

Breakpoint analysis of the RCS curve identified a risk inflection point at NLR = 1.475, indicating a change in mortality risk association. Below 1.475, NLR was not significantly linked to mortality (HR = 0.592, 95% CI: 0.343–1.022, p = 0.06). Above 1.475, each 1-unit NLR increase raised mortality risk by 13.4% (HR = 1.134, 95% CI: 1.073–1.200, p < 0.001). These findings indicate that elevated NLR beyond this breakpoint is strongly associated with increased mortality risk in the anemia population (Fig 2 and Table 4).

## Subgroup analysis

Subgroup analysis (Fig 3) showed that the association between NLR and mortality risk was consistent across most subgroups, with no significant interactions by gender, age, smoking, or drinking. However, significant interactions were observed for diabetes (HR = 1.19, 95% CI: 1.12–1.26, p = 0.01) and CHD (HR = 1.22, 95% CI: 1.11–1.34, p = 0.099), with stronger associations in these groups.

## Discussions

As expected, NLR differed significantly across pre-defined tertiles (a mathematical consequence of the grouping strategy). Thus, we focus our interpretation on non-grouping variables revealing true biological gradients. This study demonstrates that systemic inflammation, measured by the neutrophil-to-lymphocyte ratio (NLR), is strongly and independently associated with all-cause mortality in individuals with anemia. The findings suggest that NLR, a simple and widely available biomarker of systemic inflammation, could be a valuable tool for identifying high-risk individuals with anemia, particularly those with comorbid conditions such as diabetes and CHD.

The observed association between NLR and mortality aligns with previous research showing that elevated NLR is linked to adverse outcomes in various populations, including those with cardiovascular diseases [11], cancer [12,13], chronic obstructive pulmonary disease (COPD) [14], Attention-deficit hyperactivity disorder (ADHD) [15], and chronic kidney disease [16]. Elevated NLR may reflect underlying inflammatory pathways contributing to endothelial dysfunction [17]

**Table 2. Univariate logistic regression analysis of risk factors for mortality.**

| Variable | OR_95 CI | P-value |
|---|---|---|
| Age (years) | 1.11 (1.1~1.11) | <0.001 |
| Gender, n (%) | | |
| Male | 1 (reference) | |
| Female | 0.23 (0.2~0.27) | <0.001 |
| Race, n (%) | | |
| Non-Hispanic White | 1 (reference) | |
| Non-Hispanic Black | 0.3 (0.25~0.36) | <0.001 |
| Mexican American | 0.24 (0.18~0.31) | <0.001 |
| Other Hispanic | 0.25 (0.17~0.35) | <0.001 |
| Other Race – Including Multi-Racial | 0.13 (0.09~0.21) | <0.001 |
| Marital status, n (%) | | |
| Married or living with a partner | 1 (reference) | |
| Living alone | 1.35 (1.16~1.58) | <0.001 |
| Poverty Income Ratio | 0.93 (0.88~0.98) | 0.006 |
| Education, n (%) | | |
| <9th grade | 1 (reference) | |
| 9–11th grade | 0.71 (0.55~0.93) | 0.011 |
| High school | 0.57 (0.45~0.73) | <0.001 |
| College | 0.35 (0.27~0.45) | <0.001 |
| Graduate | 0.34 (0.26~0.45) | <0.001 |
| Body mass index, kg.m$^2$ | 0.97 (0.96~0.98) | <0.001 |
| Smoker, n (%) | | |
| Never | 1 (reference) | |
| Former | 3.03 (2.54~3.61) | <0.001 |
| Now | 1.62 (1.27~2.06) | <0.001 |
| Alcohol user, n (%) | | |
| Never | 1 (reference) | |
| Former | 2.98 (2.37~3.76) | <0.001 |
| Mild | 1.06 (0.84~1.33) | 0.642 |
| Moderate | 0.43 (0.3~0.61) | <0.001 |
| Heavy | 0.55 (0.4~0.77) | 0.001 |
| WBC (10$^3$/μl) | 1.01 (1~1.02) | 0.215 |
| Lymphocyte (10$^3$/μl) | 1 (0.99~1.01) | 0.858 |
| Neutrophils (10$^3$/μl) | 1.1 (1.05~1.14) | <0.001 |
| Chronic kidney disease, n (%) | | |
| No | 1 (reference) | |
| Yes | 5.83 (4.93~6.9) | <0.001 |
| Diabetes mellitus, n (%) | | |
| No | 1 (reference) | |
| Yes | 2.35 (2~2.77) | <0.001 |
| Hypertension, n (%) | | |
| No | 1 (reference) | |
| Yes | 4.16 (3.47~5) | <0.001 |
| Stroke, n (%) | | |
| No | 1 (reference) | |
| Yes | 3.48 (2.66~4.55) | <0.001 |

*(Continued)*

**Table 2.** (Continued)

| Variable | OR_95 CI | P-value |
|---|---|---|
| Coronary heart disease, n (%) | | |
| No | 1 (reference) | |
| Yes | 4.41 (3.39~5.74) | <0.001 |
| Cancer, n (%) | | |
| No | 1 (reference) | |
| Yes | 3.52 (2.85~4.34) | <0.001 |
| Thyroid disease, n (%) | | |
| No | 1 (reference) | |
| Yes | 1.15 (0.92~1.45) | 0.226 |
| Neutrophils/Lymphocyte Ratio (NLR) | 1.44 (1.35~1.52) | <0.001 |
| Neutrophils/Lymphocyte Ratio (Tertile) | | |
| T1 | 1 (reference) | |
| T2 | 1.22 (0.99~1.51) | 0.06 |
| T3 | 2.9 (2.39~3.53) | <0.001 |

**Table 3. Multivariable cox regression analysis of NLR and all-cause mortality.**

| Variable | Crude | | Model 1 | | Model 2 | | Model 3 | |
|---|---|---|---|---|---|---|---|---|
| | HR (95%CI) | P-value | HR (95%CI) | P-value | HR (95%CI) | P-value | HR (95%CI) | P-value |
| Continuous | 1.21 (1.18~1.23) | <0.001 | 1.12 (1.08~1.16) | <0.001 | 1.11 (1.07~1.15) | <0.001 | 1.11 (1.07~1.15) | <0.001 |
| Tertile | | | | | | | | |
| T1 | 0.87 (0.72~1.04) | 0.134 | 1.09 (0.9~1.32) | 0.359 | 1.13 (0.94~1.37) | 0.2 | 1.12 (0.93~1.36) | 0.229 |
| T2 | 1(Ref) | | 1(Ref) | | 1(Ref) | | 1(Ref) | |
| T3 | 2.12 (1.81~2.48) | <0.001 | 1.3 (1.11~1.53) | 0.001 | 1.26 (1.07~1.48) | 0.006 | 1.25 (1.07~1.48) | 0.007 |

and atherosclerosis [18], potentially exacerbating mortality risk [19]. Our study extends these findings to individuals with anemia, suggesting that inflammation may exacerbate the risk of poor outcomes in this population.

In our cohort, Group 3, which had the highest NLR, also had the highest prevalence of comorbidities (CKD, diabetes, hypertension, stroke, and CHD) and exhibited significantly lower rates of health-protective behaviors (never-smoking: 54.6% vs 65.1% in Group 1; never-drinking: 16.4% vs 23.1%, both p<0.001). This dual pattern may reflect two phenomena: a) Behavioral confounding: Accumulated smoking/drinking exposure in high-NLR individuals promotes inflammation; b) Sick quitter effect: Severe comorbidities may force cessation, paradoxically reducing current user counts despite prior heavy use [20]. This highlights the interplay between anemia and chronic diseases, both of which contribute to an elevated inflammatory response. Previous studies have also shown that chronic conditions can amplify systemic inflammation, which is reflected by elevated NLR [21]. The significant difference in mortality rates across the NLR tertiles further supports the hypothesis that NLR may serve as a marker of disease severity and a prognostic tool for mortality risk. Specifically, participants in the highest NLR tertile had a 25% higher risk of mortality compared to those in lower tertiles, even after adjusting for key confounders. Similar findings have been reported in other studies; for example, Smith et al. demonstrated that elevated NLR was associated with increased mortality in patients with cardiovascular diseases, independent of other inflammatory markers [22]. Their research emphasized the utility of NLR in identifying high-risk individuals due to systemic inflammation. These consistent results reinforce the role of NLR as a universal marker of inflammation and mortality risk.

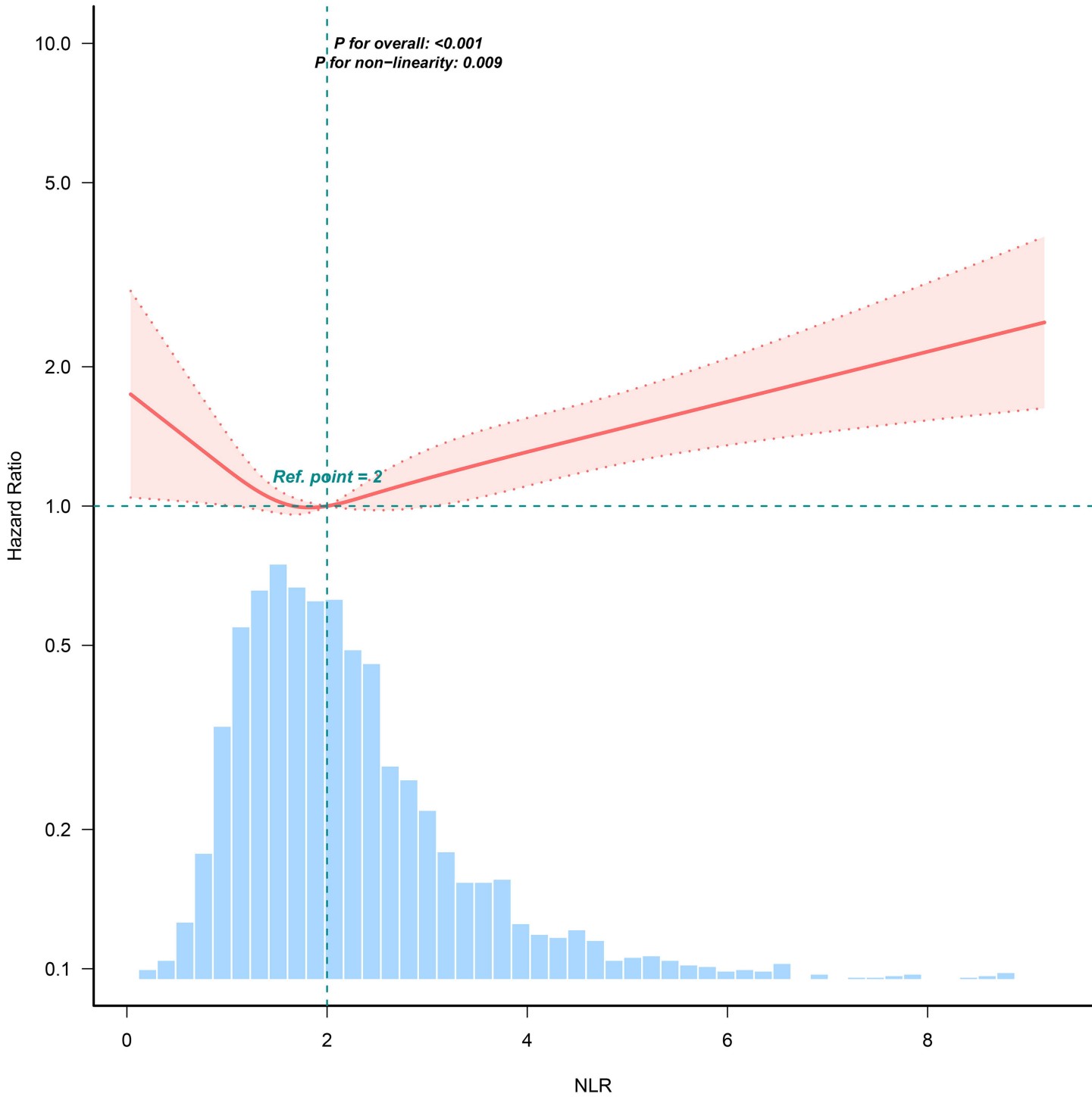

**Fig 2. Non-linear regression of NLR and mortality risk.**

**Table 4. Breakpoint analysis of NLR and mortality risk.**

| Neutrophils/Lymphocyte Ratio | Breakpoint.HR (95%CI) | P value |
|---|---|---|
| <1.47 | 0.592 (0.343,1.022) | 0.06 |
| ≥1.47 | 1.134 (1.073,1.2) | < 0.001 |
| Likelihood ratio test | | 0.025 |

While our identified NLR threshold of 1.475 is lower than thresholds reported in other populations (e.g., NLR > 3.0 in critical illness [23], > 5.0 in some cancers [24]), this likely reflects the unique vulnerability of anemic individuals to inflammation-driven morbidity. No prior study has validated this specific threshold in an anemia cohort, though similar non-linear relationships exist in CKD (NLR > 2.2) and heart failure (NLR > 3.8) [25]. The lower threshold here suggests that anemia potentiates mortality risk at milder inflammatory levels, necessitating validation in external anemia cohorts. It is important to note that this threshold (NLR = 1.475) is a data-derived inflection point from our analysis and requires validation in external and prospective cohorts of anemic individuals before clinical application. Similar thresholds have been reported in studies examining other populations, further supporting the relevance of this biomarker across different clinical contexts [26].

Subgroup analyses revealed that the association between NLR and mortality was consistent across most subgroups, including gender, age, smoking, and alcohol use. However, significant interactions were observed for diabetes and CHD. The stronger association between NLR and mortality in individuals with these conditions suggests that NLR may have particular relevance in patients with underlying chronic diseases, where inflammation plays a crucial role in disease progression and mortality risk. Prior research has similarly highlighted the role of systemic inflammation in amplifying mortality risks in patients with diabetes and CHD, reinforcing the importance of monitoring inflammatory markers in these populations [27]. These findings emphasize the need for targeted clinical interventions in individuals with anemia and comorbid conditions, particularly those with diabetes and CHD, who may be at an even higher risk of mortality.

Interpretation of our findings should consider the inherent limitations of observational designs. Using the GRADE framework for prognostic evidence [1], we classify the certainty of evidence for NLR's mortality-predictive ability as Low to Moderate (downgraded due to risk of residual confounding inherent to observational studies). Applying the Oxford Centre for Evidence-Based Medicine (CEBM) 2011 Levels of Evidence [2], this study qualifies as Level 2 evidence (prognostic cohort study with good follow-up). While the large nationally representative sample and rigorous adjustment for key confounders strengthen validity, the unavoidable indirectness of real-world data (e.g., unmeasured socioeconomic factors) and absence of external validation temper definitive causal claims. These considerations position NLR as a clinically useful risk-stratification tool rather than a definitive causal biomarker, warranting validation in intervention studies.

This study is not without its limitations. First, while the NHANES dataset is comprehensive, its retrospective cohort design limits causal inference despite longitudinal mortality linkage. Second, while NLR is a marker of systemic inflammation, it does not provide information on the specific mechanisms driving inflammation in individual patients. Future studies exploring the underlying causes of elevated NLR in individuals with anemia, as well as its potential role in guiding therapeutic interventions, are needed. Additionally, prospective studies are warranted to confirm these findings and assess whether NLR can be incorporated into clinical practice as a reliable predictor of mortality in anemia. Fourth, socioeconomic factors profoundly influence both anemia risk and systemic inflammation. While our models adjusted for key indicators (Poverty Income Ratio, education level, marital status), unmeasured factors (e.g., healthcare access, dietary quality) may contribute to residual confounding. Future studies should incorporate geographic deprivation indices or insurance status data to better quantify this relationship. Fifth, as appropriately noted by the reviewer, NHANES deliberately excludes institutionalized populations (e.g., nursing home residents, hospitalized patients) and those too ill to participate in mobile examination centers. This sampling framework necessarily underrepresents individuals with severe

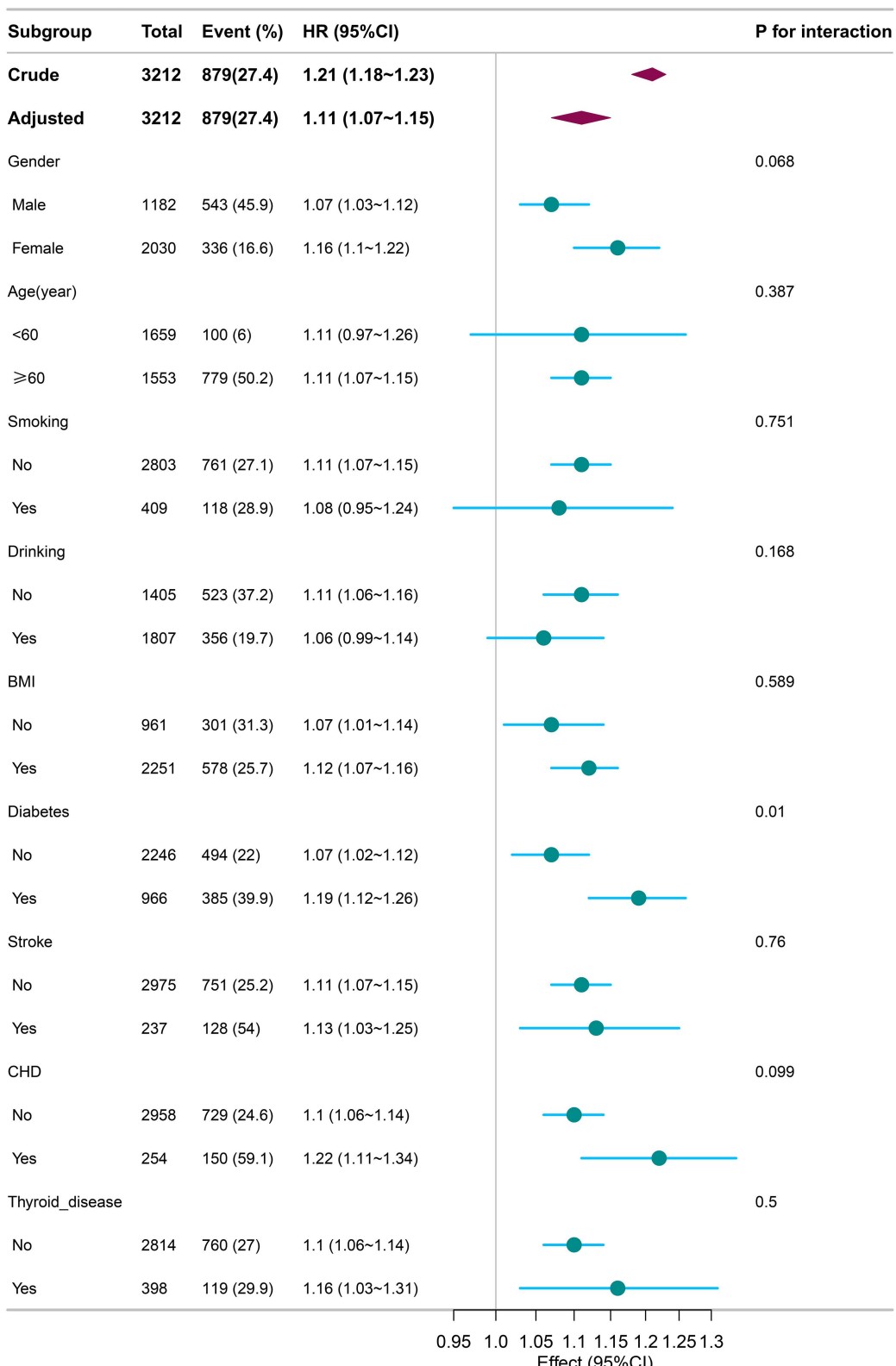

| Subgroup | Total | Event (%) | HR (95%CI) | P for interaction |
|---|---|---|---|---|
| **Crude** | **3212** | **879(27.4)** | **1.21 (1.18~1.23)** | |
| **Adjusted** | **3212** | **879(27.4)** | **1.11 (1.07~1.15)** | |
| Gender | | | | 0.068 |
| Male | 1182 | 543 (45.9) | 1.07 (1.03~1.12) | |
| Female | 2030 | 336 (16.6) | 1.16 (1.1~1.22) | |
| Age(year) | | | | 0.387 |
| <60 | 1659 | 100 (6) | 1.11 (0.97~1.26) | |
| ≥60 | 1553 | 779 (50.2) | 1.11 (1.07~1.15) | |
| Smoking | | | | 0.751 |
| No | 2803 | 761 (27.1) | 1.11 (1.07~1.15) | |
| Yes | 409 | 118 (28.9) | 1.08 (0.95~1.24) | |
| Drinking | | | | 0.168 |
| No | 1405 | 523 (37.2) | 1.11 (1.06~1.16) | |
| Yes | 1807 | 356 (19.7) | 1.06 (0.99~1.14) | |
| BMI | | | | 0.589 |
| No | 961 | 301 (31.3) | 1.07 (1.01~1.14) | |
| Yes | 2251 | 578 (25.7) | 1.12 (1.07~1.16) | |
| Diabetes | | | | 0.01 |
| No | 2246 | 494 (22) | 1.07 (1.02~1.12) | |
| Yes | 966 | 385 (39.9) | 1.19 (1.12~1.26) | |
| Stroke | | | | 0.76 |
| No | 2975 | 751 (25.2) | 1.11 (1.07~1.15) | |
| Yes | 237 | 128 (54) | 1.13 (1.03~1.25) | |
| CHD | | | | 0.099 |
| No | 2958 | 729 (24.6) | 1.1 (1.06~1.14) | |
| Yes | 254 | 150 (59.1) | 1.22 (1.11~1.34) | |
| Thyroid_disease | | | | 0.5 |
| No | 2814 | 760 (27) | 1.1 (1.06~1.14) | |
| Yes | 398 | 119 (29.9) | 1.16 (1.03~1.31) | |

0.95 1.0 1.05 1.1 1.15 1.2 1.25 1.3
Effect (95%CI)

**Fig 3. Subgroup analysis of the association between NLR and mortality risk.**

comorbidities or functional limitations who may exhibit both higher NLR values and mortality risk. Furthermore, while we adjusted for a wide range of confounders, the possibility of residual confounding (e.g., by unmeasured socioeconomic or subclinical factors) inherent in observational studies remains, which precludes definitive causal inferences. Sixth, the use of complete-case analysis for handling missing data, while common, may introduce selection bias, and the robustness of our findings should be interpreted in this context. Seventh, While causality cannot be inferred, the anemia-specific threshold and effect modification by anemia status support the biological plausibility of anemia potentiating NLR-related risk. As an observational study, our analyses identify associations rather than causal mechanisms.

In conclusion, elevated NLR is a strong and independent predictor of all-cause mortality in individuals with anemia, particularly those with comorbid conditions such as diabetes and CHD. The identification of a critical NLR threshold further strengthens its potential as a useful biomarker in clinical settings. Monitoring NLR could help clinicians better identify high-risk patients and improve management strategies, particularly in populations with anemia and underlying chronic diseases.

Future research should prioritize directly comparing the predictive performance of NLR between anemic and non-anemic populations to elucidate the potential effect modification by anemia status and further validate its specific utility in this high-risk group. Additionally, future studies should: a) Validating NLR thresholds in institutionalized anemic populations (e.g., long-term care cohorts). b) Examining whether NLR-guided interventions (e.g., nutritional support, comorbidity optimization) improve outcomes in high-risk community-dwelling adults. c) Developing integrated prediction models combining NLR with frailty metrics for vulnerable subgroups. While our internal validation supports the robustness of the NLR = 1.475 threshold, we concur that external validation is essential for clinical implementation. We propose three targeted next steps: a) Validation in disease-specific registries: Testing the threshold in curated cohorts of anemic patients with heart failure (e.g., American Heart Association Get With The Guidelines-HF) or CKD (e.g., CRIC study) where NLR may have particular pathophysiological relevance. b) Development of a composite score: Creating an integrated mortality risk score combining NLR with readily available clinical variables from our models (age, CKD status, albumin) using machine learning approaches. c) Biomarker synergy exploration: Investigating interactions between NLR and emerging inflammatory biomarkers (e.g., GDF-15, IL-6) using stored NHANES biospecimens.

## Supporting information

**S1 Fig. Receiver Operating Characteristic (ROC) curve for the univariate logistic regression model.**
(PNG)

## Author contributions

**Conceptualization:** Xiangkuan Cheng.

**Formal analysis:** Xiangkuan Cheng, Hong Zhang.

**Methodology:** Yueming Tian.

**Project administration:** Lanling Liu.

**Software:** Lanling Liu, Mingdeng Wang.

**Supervision:** yuansheng lin.

**Validation:** Hong Zhang, Mingdeng Wang, yuansheng lin.

**Visualization:** Mingdeng Wang.

**Writing – original draft:** Yueming Tian, Hong Zhang.

**Writing – review & editing:** yuansheng lin.

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
