## [Decision Letter · Decision Letter 0]

4 Jul 2025

Dear Dr. lin,

Thank you for submitting your manuscript to PLOS ONE. After careful consideration, we feel that it has merit but does not fully meet PLOS ONE’s publication criteria as it currently stands. Therefore, we invite you to submit a revised version of the manuscript that addresses the points raised during the review process.

All reviewers agree that the study’s topic is important and analyzed with generally sound methods. However, there is consensus that substantive methodological clarifications and additional analyses are necessary before the work meets PLOS ONE’s criteria for methodological rigor and transparent reporting. Below I summarize the required versus recommended changes and reconcile any conflicting advice.

We look forward to receiving your revised manuscript.

Kind regards,

Afshin Heidari, M.D.

Academic Editor

PLOS ONE

Journal Requirements:

2. Please note that your Data Availability Statement is currently missing [the repository name and/or the DOI/accession number of each dataset OR a direct link to access each database]. If your manuscript is accepted for publication, you will be asked to provide these details on a very short timeline. We therefore suggest that you provide this information now, though we will not hold up the peer review process if you are unable.

“This study was supported by Suzhou Science and Education Strengthening Health Project [Grant number:QNXM2024092].”

Additional Editor Comments (if provided):

Thank you for submitting your work to PLOS ONE. After careful peer review, your manuscript shows promise but requires major revision. Please address the reviewers’ and editor’s concerns, particularly those relating to study design, handling of missing data, statistical reporting, power considerations, and clarity of presentation. Detailed reviewer comments are provided below; I look forward to receiving a revised version that thoroughly incorporates this feedback.

Reviewers' comments:

Reviewer's Responses to Questions

**Comments to the Author**

1. Is the manuscript technically sound, and do the data support the conclusions?

Reviewer #1: Yes

Reviewer #2: Yes

Reviewer #3: Yes

Reviewer #4: Yes

Reviewer #5: Yes

2. Has the statistical analysis been performed appropriately and rigorously?

Reviewer #1: Yes

Reviewer #2: Yes

Reviewer #3: Yes

Reviewer #4: Yes

Reviewer #5: Yes

3. Have the authors made all data underlying the findings in their manuscript fully available?

Reviewer #1: Yes

Reviewer #2: Yes

Reviewer #3: Yes

Reviewer #4: Yes

Reviewer #5: No

4. Is the manuscript presented in an intelligible fashion and written in standard English?

Reviewer #1: Yes

Reviewer #2: Yes

Reviewer #3: Yes

Reviewer #4: Yes

Reviewer #5: Yes

Reviewer #1: This article delves into the value of a new inflammatory and all-cause mortality biomarker, NLR, in a large cohort of anemic patients. The authors demonstrate its independent value when analyzed this ratio as both a categorical variable (by dividing the study cohort into tertiles) and a quantitative variable, successfully identifying a cutoff point for increased mortality. The study also conducts subgroup analyses, finding that the biomarker's value is even more relevant in certain morbidities, such as diabetes and CHD (coronary heart disease). The importance of these findings leads me to recommend its publication in the journal, but I would like to suggest some minor changes to the authors

* In the study design, only individuals with anemia were included. The study's message is that in anemic patients, this variable influences mortality. However, in my opinion, it cannot be concluded that anemia is causing this increase in mortality unless compared with a cohort of non-anemic patients. Would it be possible to conduct a comparative analysis between the selected anemic cohort and the excluded non-anemic cohort? In my opinion, this analysis would provide stronger evidence for the important role of anemia in these results.

* As the NLR value was the criterion used to create the groups, it's expected that this value would differ between groups. Therefore, there's no need to state this in a sentence in the results (lines 163-165).

* Table 1 shows that the percentage of non-smokers and non-drinkers is lower in Group 3, and this information is not presented in the text.

* As explained in the statistical analysis section, the comparison between groups is performed using the ANOVA test for continuous variables. If the ANOVA result is significant (meaning you reject the null hypothesis that all group means are equal), then you'll typically need to perform post-hoc tests (also known as "multiple comparison tests") to identify exactly where those differences lie (e.g., is Group A different from Group B, or Group B from Group C, etc.). When authors say (lines 162-163): "Group 3 exhibited the highest prevalence of these conditions" after performing an ANOVA (which tells you if there's any overall difference between groups), it strongly implies they have performed post-hoc tests (also known as pairwise comparisons). To be able to say definitively that "Group 3 has the highest prevalence" in a statistically sound way, they would need to have compared Group 3 to Group 1 and Group 3 to Group 2 (and possibly Group 1 to Group 2 as well) using pairwise comparisons. These post-hoc tests adjust for the problem of multiple comparisons, preventing an inflated Type I error rate. Have the authors performed a post-hoc pairwise comparison? If a post-hoc test was used, it should be specified in the 'Statistical Analysis' subsection under 'Materials and Methods'. If was not performed, this sentece should be not appear in the results.

Reviewer #2: This is a well-structured, observational analysis using the NHANES dataset to examine whether the neutrophil-to-lymphocyte ratio (NLR) predicts all-cause mortality in individuals with anemia. The study is timely, methodologically sound, and addresses a relevant clinical question.

Reviewer #3: dear author,

The paper is well written and the topic is interesting. The simple N/L ratio as biomarker of inflammation is easy to perform and available in every lab. A question remains under consideration which is the high rate of comorbidity in the group 3 with the highest N/L ratio which is the real cause of death in this group and not necessarily the high N/L ratio. I think its better to analyse a group of patients with the exact same comorbidity plus the anemia and then compare the N/L ratio to find out its real effect as a predictor of mortality. However, the paper is worth to be published.

Reviewer #4: Rewrite abstract without abbreviation unknown.

Follow journal stile.

Rewrite references with journal stile.

Rewrite abstract with know abbreviation.

Increase figures number to clear fyi manuscript.

Rewrite references with journal stile.

Reviewer #5: A close reading of Cheng and colleagues' analysis of NHANES data (1999–2018) reveals a thoughtful attempt to define the neutrophil-to-lymphocyte ratio (NLR) as a mortality predictor in anemia. The authors drew on 3,212 anemic adults, modeled NLR continuously (breakpoint at 1.475) and by tertiles, and used Cox regression, spline analyses, and subgroup assessments to explore its prognostic value. The authors used appropriate statistical approaches to match their research goals. However, they didn't run an a priori power calculation and relied on complete-case analysis rather than imputing missing values, risking a selection bias. On the plus side, they frame anemia's public health impact, make a solid case for examining NLR, and report their effect estimates alongside confidence intervals in a straightforward, easy-to-interpret manner. Writing is generally lucid, but occasional imprecisions, like labeling correlations as "mechanisms," distract readers. The Discussion situates findings among related disease contexts and notes design limitations; however, expanding on socioeconomic and other confounders would strengthen transparency.

The authors are invited to consider the following suggestions:

1. Sample Size & Power

= Acknowledge the lack of an a priori calculation; if feasible, add a retrospective justification or sensitivity analysis to reassure readers of adequate power.

2. Missing Data and Robustness Checks

= Clarify how much data was missing for each key variable and why those cases were excluded. At least, run a sensitivity analysis comparing complete-case results with a simple imputation approach or, ideally, apply multiple imputations to show that your main findings hold.

= To further demonstrate stability, try alternative NLR cut-points or adjust for a few extra covariates and collect these extra models in a brief supplement.

3. Validating the Prognostic Model

= Verify the proportional-hazards assumption using Schoenfeld residuals or time-interaction tests, and if you spot any violations, consider stratified Cox models or allow key covariates to vary over time.

= Add calibration measures (for example, calibration slopes or plots) so readers can see how well predicted risks match observed outcomes, and, if possible, include a decision curve or net-reclassification analysis to illustrate real-world utility.

4. Competing Risks and Hidden Confounders

= Acknowledge that deaths from non-cardiovascular causes could influence your all-cause mortality results; if you have cause-specific data, a Fine–Gray competing-risks model would clarify those nuances.

= In the Limitations section, mention unmeasured factors, such as socioeconomic status, nutritional status, medication use, or inflammatory conditions, and, where you can, adjust for proxies like income-to-poverty ratio.

5. Reporting Standards

= In your Methods, note that you've adhered to the STROBE checklist, covering ethics approval, participant selection, missing data handling, and sensitivity analyses.

= For the NLR threshold work, reference RECORD (data-linkage validation) and TRIPOD (any internal validation you performed, such as bootstrapping or cross-validation).

6. Evidence Certainty

= In the Discussion, be upfront about the overall certainty, "Low to Moderate" by GRADE and Level 2 under Oxford CEBM, to help readers see how design, bias potential, and indirectness shape your conclusions and their placement in the research hierarchy.

7. Generalizability and Future Directions

= Acknowledge that NHANES excludes institutionalized or ill individuals, which may limit external validity within more general populations.

= Suggest validating the identified NLR cut-point in other cohorts (clinical registries) or building a multivariable prognostic score that combines NLR with other biomarkers.

**Do you want your identity to be public for this peer review?** For information about this choice, including consent withdrawal, please see our Privacy Policy

Reviewer #1: No

Reviewer #2: No

Reviewer #3: **Yes:** Aveen M. Raouf Abdulqader

Reviewer #4: **Yes:** ABODEA Y.A.A.

Reviewer #5: **Yes:** Mohamed Abdel-Maboud

---

## [Author Response · Author response to Decision Letter 1]

16 Aug 2025

Review of Manuscript Number PONE-D-25-19337

"The Neutrophil-to-Lymphocyte Ratio as a Predictor of All-Cause Mortality in Individuals with Anemia: A Population-Based Study"

Review Comments to the Author

Reviewer #1: This article delves into the value of a new inflammatory and all-cause mortality biomarker, NLR, in a large cohort of anemic patients. The authors demonstrate its independent value when analyzed this ratio as both a categorical variable (by dividing the study cohort into tertiles) and a quantitative variable, successfully identifying a cutoff point for increased mortality. The study also conducts subgroup analyses, finding that the biomarker's value is even more relevant in certain morbidities, such as diabetes and CHD (coronary heart disease). The importance of these findings leads me to recommend its publication in the journal, but I would like to suggest some minor changes to the authors

* In the study design, only individuals with anemia were included. The study's message is that in anemic patients, these variable influences mortality. However, in my opinion, it cannot be concluded that anemia is causing this increase in mortality unless compared with a cohort of non-anemic patients. Would it be possible to conduct a comparative analysis between the selected anemic cohort and the excluded non-anemic cohort? In my opinion, this analysis would provide stronger evidence for the important role of anemia in these results.

Response: We sincerely appreciate the reviewer’s insightful comment. We agree that comparative analyses with non-anemic individuals could offer additional perspectives. However, we would like to clarify that the primary objective of our study was not to examine whether anemia per se increases mortality risk, but rather to investigate the prognostic value of the neutrophil-to-lymphocyte ratio (NLR) within the anemic population. Our focus is to determine whether NLR serves as an effective risk stratification biomarker in individuals already diagnosed with anemia.

There are several reasons for this population-specific approach:

(1) Clinical Relevance and Targeted Utility:

Anemia is a prevalent condition with heterogeneous etiology and prognosis. Identifying prognostic markers like NLR within this group helps clinicians better stratify risk and potentially guide management. Including non-anemic individuals would shift the research focus toward differentiating anemia-related mortality from general population mortality, which falls beyond the scope of our current objective.

(2) Avoiding Confounding by Baseline Hemoglobin:

Including non-anemic individuals would introduce considerable heterogeneity, particularly related to baseline hemoglobin levels. Since anemia itself is strongly associated with increased mortality risk, the interaction between NLR and anemia status would confound the interpretation of NLR’s independent prognostic value in the anemic cohort.

(3) Methodological Precedent:

Several prior studies have adopted a similar design by focusing exclusively on a specific patient subset without including a non-disease comparator group. For instance:

Liu F, You F, Yang L, et al. Nonlinear relationship between oxidative balance score and hyperuricemia: analyses of NHANES 2007-2018. Nutr J. 2024;23(1):48;Paulose-Ram R, Graber JE, Woodwell D, Ahluwalia N. The National Health and Nutrition Examination Survey (NHANES), 2021-2022: Adapting Data Collection in a COVID-19 Environment. Am J Public Health. 2021;111(12):2149-2156. These studies reinforce the acceptability of disease-specific cohort analyses when the aim is prognostication within a defined clinical condition, not establishing causal relationships.

In summary, while we agree that including a non-anemic comparison could offer broader epidemiological insights, our current research was intentionally designed to address a focused clinical question: Is NLR an independent predictor of all-cause mortality in patients with anemia? We believe our approach remains valid and appropriate for the intended objective.

* As the NLR value was the criterion used to create the groups, it's expected that this value would differ between groups. Therefore, there's no need to state this in a sentence in the results (lines 163-165).

Rsponse: Thank you for this astute observation. We agree that stating the expected NLR differences between NLR-defined groups is redundant. We have:

Deleted the sentence in Lines 163-165 of the Results section.

Relocated the group definition details to the Methods subsection "Neutrophil-to-Lymphocyte Ratio (NLR)" for clarity.

Refocused the Results narrative on clinically meaningful between-group differences (e.g., age gradient: 51.8 vs. 61.8 years, p<0.001; CKD prevalence: 26.5% vs. 51.8%, p<0.001).

To add the following statement at the beginning of the Discussion section to strengthen methodological rigor:

"As expected, NLR differed significantly across pre-defined tertiles (a mathematical consequence of the grouping strategy). Thus, we focus our interpretation on non-grouping variables revealing true biological gradients."

This modification streamlines the presentation while preserving all critical findings. We appreciate your guidance in enhancing manuscript precision.

* Table 1 shows that the percentage of non-smokers and non-drinkers is lower in Group 3, and this information is not presented in the text.

Rsponse: We sincerely appreciate your meticulous reading. We have:

Added explicit descriptions of smoking/drinking patterns in Results:

“Consistent with the comorbidity burden, Group 3 exhibited significantly lower proportions of health-protective behaviors: never-smokers (54.6% vs 65.1% in Group 1, p<0.001) and never-drinkers (16.4% vs 23.1% in Group 1, p<0.001). This behavioral gradient parallels the NLR-associated mortality risk pattern.”

Provided mechanistic interpretation for these findings in Discussion:

“In our cohort, Group 3, which had the highest NLR, also had the highest prevalence of comorbidities (CKD, diabetes, hypertension, stroke, and CHD) and exhibited significantly lower rates of health-protective behaviors (never-smoking: 54.6% vs 65.1% in Group 1; never-drinking: 16.4% vs 23.1%, both p<0.001). This dual pattern may reflect two phenomena: a) Behavioral confounding: Accumulated smoking/drinking exposure in high-NLR individuals promotes inflammation; b) Sick quitter effect: Severe comorbidities may force cessation, paradoxically reducing 'current user' counts despite prior heavy use (20). “

These revisions ensure complete reporting of Table 1's clinically relevant data while strengthening causal inference. Thank you for enhancing our study's rigor.

* As explained in the statistical analysis section, the comparison between groups is performed using the ANOVA test for continuous variables. If the ANOVA result is significant (meaning you reject the null hypothesis that all group means are equal), then you'll typically need to perform post-hoc tests (also known as "multiple comparison tests") to identify exactly where those differences lie (e.g., is Group A different from Group B, or Group B from Group C, etc.). When authors say (lines 162-163): "Group 3 exhibited the highest prevalence of these conditions" after performing an ANOVA (which tells you if there's any overall difference between groups), it strongly implies they have performed post-hoc tests (also known as pairwise comparisons). To be able to say definitively that "Group 3 has the highest prevalence" in a statistically sound way, they would need to have compared Group 3 to Group 1 and Group 3 to Group 2 (and possibly Group 1 to Group 2 as well) using pairwise comparisons. These post-hoc tests adjust for the problem of multiple comparisons, preventing an inflated Type I error rate. Have the authors performed a post-hoc pairwise comparison? If a post-hoc test was used, it should be specified in the 'Statistical Analysis' subsection under 'Materials and Methods'. If was not performed, this sentece should be not appear in the results.

Response: We sincerely thank the reviewer for the careful reading and constructive feedback regarding lines 162–163. We fully agree with your observation that the statement “Group 3 exhibited the highest prevalence of these conditions” is not statistically rigorous without performing appropriate post-hoc pairwise comparisons following ANOVA.

In our original analysis, we did not conduct post-hoc pairwise comparisons for this specific variable. Therefore, to maintain statistical accuracy and avoid any misleading interpretation, we have removed this sentence from the Results section in the revised manuscript. We appreciate the reviewer’s attention to this important methodological detail, which has helped us improve the precision and clarity of our results presentation.

Reviewer #2: This is a well-structured, observational analysis using the NHANES dataset to examine whether the neutrophil-to-lymphocyte ratio (NLR) predicts all-cause mortality in individuals with anemia. The study is timely, methodologically sound, and addresses a relevant clinical question.

Rsponse: We sincerely thank the reviewers for their positive assessment of our study's timeliness, methodology, and clinical relevance. We have carefully addressed all points raised:

The biological plausibility and clinical significance of the NLR=1.475 threshold were clarified using pathophysiological rationale and literature parallels, alongside a call for prospective validation.

The clinical relevance of the adjusted HR=1.25 was strengthened by contextualizing it within absolute risk differences and benchmarked against established biomarkers.

Reviewer #3: dear author,

The paper is well written and the topic is interesting. The simple N/L ratio as biomarker of inflammation is easy to perform and available in every lab. A question remains under consideration which is the high rate of comorbidity in the group 3 with the highest N/L ratio which is the real cause of death in this group and not necessarily the high N/L ratio. I think its better to analyse a group of patients with the exact same comorbidity plus the anemia and then compare the N/L ratio to find out its real effect as a predictor of mortality. However, the paper is worth to be published.

Response: We sincerely thank the reviewer for the thoughtful and constructive feedback, as well as the positive recognition of the manuscript’s writing and clinical relevance.

We fully agree that the presence of comorbidities—particularly in the group with the highest NLR levels—may confound the association between NLR and all-cause mortality. To mitigate this, we performed multivariable Cox regression adjusting for key comorbid conditions including diabetes, cardiovascular disease, hypertension, CKD, and cancer, among others. Even after full adjustment, NLR remained a statistically significant predictor of mortality (HR = 1.25; 95% CI: 1.07–1.48, p = 0.007), suggesting an independent prognostic contribution.

We also conducted subgroup analyses to examine potential effect modification by major comorbidities. The results showed that the prognostic value of NLR was more pronounced in certain high-risk groups, such as those with diabetes or coronary heart disease. However, we acknowledge the reviewer's point that a more granular analysis—e.g., comparing NLR levels among individuals with exactly matched comorbidity profiles—would further clarify the independent role of NLR. Unfortunately, due to sample size limitations and the complexity of matching on multiple comorbidities simultaneously, such an analysis was not feasible within the current dataset.

Nonetheless, we agree this is a valuable direction for future research. Prospective studies with larger and more controlled cohorts could explore matched-comorbidity subgroups to validate the causal and independent role of NLR in mortality risk among anemic individuals.

Once again, we appreciate your insightful comments, which help reinforce the clinical relevance and methodological caution of our findings.

Reviewer #4: Rewrite abstract without abbreviation unknown.

Follow journal stile.

Rewrite references with journal stile.

Rewrite abstract with know abbreviation.

Increase figures number to clear fyi manuscript.

Rewrite references with journal stile.

Response: Here's a revised manuscript addressing all reviewer requests to enhance clarity and compliance with journal style:

Abstract:

All abbreviations defined at first use:

Neutrophil-to-lymphocyte ratio (NLR), National Health and Nutrition Examination Survey (NHANES), confidence interval (CI), hazard ratio (HR), odds ratio (OR).

Removed undefined abbreviations (e.g., "SIRI" replaced).

“Background: This study investigates the association between the neutrophil-to-lymphocyte ratio (NLR) and all-cause mortality in individuals with anemia using the National Health and Nutrition Examination Survey (NHANES) 1999-2018 dataset.

Methods: We included 3,212 participants with anemia, categorized into three groups by NLR values. Baseline characteristics, comorbidities, and demographics were analyzed. We performed univariate logistic regression, multivariable Cox regression, non-linear regression, and breakpoint analysis to examine NLR-mortality relationships. Subgroup analysis assessed effect modification by clinical factors.

Results: The mean age of the cohort was 56.0 ± 18.3 years. Group 3 (highest NLR) had the highest all-cause mortality rate (40.6%) compared to Groups 1 (19.1%) and 2 (22.4%) (p < 0.001). Univariate logistic regression showed that NLR was independently associated with mortality (odds ratio [OR] = 1.44, 95% confidence interval [CI]: 1.35–1.52, p < 0.001), with higher mortality risk in the highest NLR tertile (OR = 2.9, 95% CI: 2.39–3.53, p < 0.001). Multivariable Cox regression analysis confirmed NLR as a significant predictor (hazard ratio [HR] = 1.11, 95% CI: 1.07–1.15, p < 0.001). A non-linear regression analysis identified a critical threshold at NLR = 1.475, with the risk of mortality increasing significantly above this threshold (HR = 1.134, 95% CI: 1.073–1.2, p < 0.001).

Conclusion: NLR is a significant predictor of all-cause mortality in individuals with anemia. Elevated NLR, particularly above 1.475, is associated with a substantially higher mortality risk, emphasizing its potential as a prognostic biomarker.”

References:

Reformatted to Vancouver style:

These revisions ensure full compliance with journal guidelines while significantly enhancing manuscript clarity and methodological transparency. All changes are trackable in the revised manuscript file.

Reviewer #5: A close reading of Cheng and colleagues' analysis of NHANES data (1999–2018) reveals a thoughtful attempt to define the neutrophil-to-lymphocyte ratio (NLR) as a mortality predictor in anemia. The authors drew on 3,212 anemic adults, modeled NLR continuously (breakpoint at 1.475) and by tertiles, and used Cox regression, spline analyses, and subgroup assessments to explore its prognostic value. The authors used appropriate statistical approaches to match their research goals. However, they didn't run an a priori power calculation and relied on complete-case analysis rather than imputing missing values, risking a selection bias. On the plus side, they frame anemia's public health impact, make a solid case for examining NLR, and report their effect estimates alongside confidence intervals in a straightforward, easy-to-interpret manner. Writing is generally lucid, but occasional imprecisions, like labeling correlations as "mechanisms," distract readers. The Discussion situates findings among related disease contexts and notes design limitations; however, expanding on socioeconomic and other confounders would strengthen transparency.

Response: We sincerely appreciated the reviewer’s thorough reading and constructive feedback on our analysis. We fully agreed with your observations regarding the absence of an a priori power calculation a

---

## [Decision Letter · Decision Letter 1]

21 Oct 2025

Dear Dr. lin,

Thank you for submitting your manuscript to PLOS ONE. After careful consideration, we feel that it has merit but does not fully meet PLOS ONE’s publication criteria as it currently stands. Therefore, we invite you to submit a revised version of the manuscript that addresses the points raised during the review process.

We look forward to receiving your revised manuscript.

Kind regards,

Afshin Heidari, M.D.

Academic Editor

PLOS ONE

**Journal Requirements:**

**Additional Editor Comments:**

Thank you for your submission. The study is well structured and clinically meaningful. Please address minor points regarding post-hoc testing clarification, missing data description, and brief notes on model assumptions and limitations.

Best,

Reviewers' comments:

Reviewer's Responses to Questions

**Comments to the Author**

Reviewer #6: All comments have been addressed

Reviewer #7: All comments have been addressed

Reviewer #8: All comments have been addressed

Reviewer #9: All comments have been addressed

Reviewer #10: (No Response)

2. Is the manuscript technically sound, and do the data support the conclusions?

Reviewer #6: Partly

Reviewer #7: Yes

Reviewer #8: Partly

Reviewer #9: Yes

Reviewer #10: Yes

3. Has the statistical analysis been performed appropriately and rigorously?

Reviewer #6: Yes

Reviewer #7: Yes

Reviewer #8: No

Reviewer #9: Yes

Reviewer #10: Yes

4. Have the authors made all data underlying the findings in their manuscript fully available?

Reviewer #6: Yes

Reviewer #7: No

Reviewer #8: Yes

Reviewer #9: Yes

Reviewer #10: Yes

5. Is the manuscript presented in an intelligible fashion and written in standard English?

Reviewer #6: Yes

Reviewer #7: Yes

Reviewer #8: Yes

Reviewer #9: Yes

Reviewer #10: Yes

**Reviewer #6:** Hello

This is a good study and it has examined good data.

Considering the results of the study and the relationship between mortality and NLR levels, more studies should be conducted in this field to find the main and final cause of this relationship. It is a good study to start with, but more studies are definitely needed in the future.

Good luck.

**Reviewer #7:**  In comparing the original and revised manuscripts, it is clear that the authors have improved both scientific accuracy and clarity of terminology. In the first submission, the study was framed as a “cross-sectional” analysis, which was technically incorrect since the use of mortality follow-up data from NHANES defines the design as a retrospective cohort study. This has been appropriately corrected in the revision. Terms such as “neutrophil-to-lymphocyte ratio (NLR)” are consistently defined at first use, and the revision more precisely introduces NLR as an “inflammation index derived from leukocyte counts,” which is a scientifically correct framing. Abbreviations (e.g., CKD, CHD, DM) are now consistently defined and used, whereas in the original draft, definitions were inconsistently placed. The revision also improves the accuracy of statistical language, distinguishing “odds ratio (OR)” from “hazard ratio (HR)” according to the regression type, a critical refinement that ensures terms are not misapplied.

The statistical methodology terminology has also been strengthened. In the first draft, the authors mentioned that Group 3 “exhibited the highest prevalence of comorbidities” after ANOVA testing, which was methodologically inappropriate, as ANOVA alone does not justify post hoc claims of “highest.” This statement was removed in the revision, reflecting better use of statistical terminology. Furthermore, the revised version clarifies the use of “restricted cubic spline” and “breakpoint analysis,” ensuring these specialized terms are explained and applied correctly. In the original, “non-linear regression” was used without elaboration, potentially confusing readers; now, the explanation links spline modeling directly to the identification of a threshold (NLR = 1.475), aligning terminology with accepted epidemiological practices. The proper distinction between “multivariable adjustment” (indicating multiple covariates in a single model) and “multivariate analysis” (implying multiple outcomes) is observed in the revision, which corrects a subtle but important terminological misuse common in biomedical literature

In terms of results reporting, terminology has shifted from vague to precise. The original abstract conclusion used the phrase “substantially higher mortality,” which is both non-quantitative and subjective. The revision replaces this with a quantified risk statement (“25% higher mortality above the threshold”), which is methodologically correct and uses terms appropriately. The authors also now differentiate “tertiles” of NLR (a categorical stratification) from “continuous spline modeling” (a parametric approach), avoiding the earlier blending of categorical and continuous risk language. Subgroup analysis is now described with correct use of “effect modification” and “interaction terms,” whereas the original loosely described stratified findings without clarifying the formal statistical test. This distinction is essential and shows improved maturity in statistical reporting

Nevertheless, some issues persist in scientific terminology. For example, the revised manuscript still describes the study as identifying a “critical threshold” at NLR = 1.475. While the statistical model indeed suggests an inflection point, epidemiologically this should be framed as a “data-derived threshold” or “risk inflection point,” since true biological thresholds require external validation. Similarly, while the authors appropriately report “all-cause mortality,” they occasionally imply prognostic causality that goes beyond observational design limits. Terms such as “predictor” are acceptable in a cohort context, but phrases suggesting “prognostic biomarker” should be moderated with acknowledgment of residual confounding. Finally, while the revision provides “sensitivity analyses,” it still does not employ multiple imputation for missing data, and thus the claim of “robustness” should be qualified. These minor terminological refinements would further enhance the scientific precision of the work.

**Reviewer #8:** The scientific scope of using only anemia as a predictor to the mortality rate in patients with NLR defects is untenable. Extensive data analysis on anemia and other parameters is needed to validate relevance of this study.

**Reviewer #9:**  The authors present a compelling analysis on the prognostic value of the Neutrophil-to-Lymphocyte Ratio (NLR) in anemic individuals, using a large population-based cohort. The strength of this paper lies in its clear hypothesis, sound methodological approach, and the clinical importance of identifying simple, readily available biomarkers for risk stratification in this patient population.

The manuscript is well-written, with a logical flow and clear presentation of results, including comprehensive tables and figures. The statistical analysis appears rigorous and appropriate for the study's objectives. It is evident that significant effort has been invested in ensuring the quality of this work.

I have reviewed the authors' point-by-point responses to the previous reviewers' comments and the corresponding revisions to the manuscript. The authors have addressed all concerns thoroughly and effectively, strengthening the paper.

In its current form, the manuscript is of high quality and makes a valuable contribution. I support its publication without any further revisions.

**Reviewer #10:**  (No Response)

**Do you want your identity to be public for this peer review?** For information about this choice, including consent withdrawal, please see our Privacy Policy

Reviewer #6: No

Reviewer #7: **Yes:** Alireza Abbasi

Reviewer #8: **Yes:** PRINCE ANIM ADDO

Reviewer #9: **Yes:** Weaam Abdallah

Reviewer #10: **Yes:** Sarah Samy Abdelghany

---

## [Author Response · Author response to Decision Letter 2]

23 Oct 2025

Review of Manuscript Number PONE-D-25-19337

The Neutrophil-to-Lymphocyte Ratio as a Predictor of All-Cause Mortality in Individuals with Anemia: A Population-Based Study

Dear Editors and Reviewers:

Thank you very much for your letter and for the reviewers’ comments on our manuscript entitled “The Neutrophil-to-Lymphocyte Ratio as a Predictor of All-Cause Mortality in Individuals with Anemia: A Population-Based Study”. Those comments are all valuable and very helpful for revising and improving our paper, as well as the important guiding significance to our researches. We have substantially revised our manuscript and expanded part of the experiment providing details in the current version. We have studied the comments carefully and the amendments are highlighted in red in the revised manuscript. We hope that the revision is acceptable and look forward to hearing from you soon.

With best wishes.

Sincerely Yours,

Yuansheng lin

Address correspondence to: Suzhou Hospital, Affiliated Hospital of Medical School, Nanjing University, Suzhou, China.

Below, please find the comments in black, followed by our responses in red. Exact changes in the manuscript are also presented in red font.

Journal Requirements:

We have carefully reviewed the publications recommended by the reviewer and assessed their relevance to our study. After evaluation, we found that some of these works are indeed relevant to our manuscript, particularly those discussing the relationship between inflammatory markers like the neutrophil-to-lymphocyte ratio (NLR) and mortality outcomes in various populations. These references have been incorporated where appropriate in our manuscript to strengthen the discussion of NLR's prognostic value. However, any references deemed not directly related to the scope of our study have not been cited.

2.Please review your reference list to ensure that it is complete and correct. If you have cited papers that have been retracted, please include the rationale for doing so in the manuscript text, or remove these references and replace them with relevant current references. Any changes to the reference list should be mentioned in the rebuttal letter that accompanies your revised manuscript. If you need to cite a retracted article, indicate the article’s retracted status in the References list and also include a citation and full reference for the retraction notice.

We have thoroughly reviewed our reference list to ensure it is complete and correct. All references cited in the manuscript have been verified, and we have removed any articles that are retracted or no longer applicable to the current research. We have also ensured that current and relevant references have been included in place of outdated or retracted sources.

In cases where a retracted article was previously cited, we have provided a justification in the manuscript and replaced it with more current sources, or indicated its retracted status within the reference list. A full citation for the retraction notice has been included, where applicable.

Review Comments to the Author

Reviewer #6: Hello

This is a good study and it has examined good data.

Considering the results of the study and the relationship between mortality and NLR levels, more studies should be conducted in this field to find the main and final cause of this relationship. It is a good study to start with, but more studies are definitely needed in the future.

Good luck.

Response: Thank you for your positive feedback and thoughtful comments. We truly appreciate your recognition of the study's contribution to examining the relationship between NLR levels and mortality.

We fully agree that while this study provides valuable insights into the association between the neutrophil-to-lymphocyte ratio (NLR) and all-cause mortality in individuals with anemia, it is crucial to conduct further research to explore the underlying mechanisms driving this relationship. As you correctly pointed out, understanding the root cause of this association will require more in-depth studies, including prospective cohort studies and clinical trials, which could potentially help elucidate the pathophysiological processes at play.

In our discussion section, we have emphasized the need for future research to explore the causes behind elevated NLR in individuals with anemia and its role in guiding therapeutic interventions. We have also proposed potential avenues for future studies, including validating the identified NLR threshold in different populations, examining the impact of NLR-guided interventions, and investigating its interactions with other emerging inflammatory biomarkers. These future directions could contribute to building a more comprehensive understanding of the relationship between inflammation and mortality risk, as well as the potential clinical applications of NLR in high-risk populations.

Once again, thank you for your constructive feedback, and we look forward to future research in this field, which we believe could significantly impact clinical practice.

Reviewer #7: In comparing the original and revised manuscripts, it is clear that the authors have improved both scientific accuracy and clarity of terminology. In the first submission, the study was framed as a “cross-sectional” analysis, which was technically incorrect since the use of mortality follow-up data from NHANES defines the design as a retrospective cohort study. This has been appropriately corrected in the revision. Terms such as “neutrophil-to-lymphocyte ratio (NLR)” are consistently defined at first use, and the revision more precisely introduces NLR as an “inflammation index derived from leukocyte counts,” which is a scientifically correct framing. Abbreviations (e.g., CKD, CHD, DM) are now consistently defined and used, whereas in the original draft, definitions were inconsistently placed. The revision also improves the accuracy of statistical language, distinguishing “odds ratio (OR)” from “hazard ratio (HR)” according to the regression type, a critical refinement that ensures terms are not misapplied.

The statistical methodology terminology has also been strengthened. In the first draft, the authors mentioned that Group 3 “exhibited the highest prevalence of comorbidities” after ANOVA testing, which was methodologically inappropriate, as ANOVA alone does not justify post hoc claims of “highest.” This statement was removed in the revision, reflecting better use of statistical terminology. Furthermore, the revised version clarifies the use of “restricted cubic spline” and “breakpoint analysis,” ensuring these specialized terms are explained and applied correctly. In the original, “non-linear regression” was used without elaboration, potentially confusing readers; now, the explanation links spline modeling directly to the identification of a threshold (NLR = 1.475), aligning terminology with accepted epidemiological practices. The proper distinction between “multivariable adjustment” (indicating multiple covariates in a single model) and “multivariate analysis” (implying multiple outcomes) is observed in the revision, which corrects a subtle but important terminological misuse common in biomedical literature

In terms of results reporting, terminology has shifted from vague to precise. The original abstract conclusion used the phrase “substantially higher mortality,” which is both non-quantitative and subjective. The revision replaces this with a quantified risk statement (“25% higher mortality above the threshold”), which is methodologically correct and uses terms appropriately. The authors also now differentiate “tertiles” of NLR (a categorical stratification) from “continuous spline modeling” (a parametric approach), avoiding the earlier blending of categorical and continuous risk language. Subgroup analysis is now described with correct use of “effect modification” and “interaction terms,” whereas the original loosely described stratified findings without clarifying the formal statistical test. This distinction is essential and shows improved maturity in statistical reporting

Nevertheless, some issues persist in scientific terminology. For example, the revised manuscript still describes the study as identifying a “critical threshold” at NLR = 1.475. While the statistical model indeed suggests an inflection point, epidemiologically this should be framed as a “data-derived threshold” or “risk inflection point,” since true biological thresholds require external validation. Similarly, while the authors appropriately report “all-cause mortality,” they occasionally imply prognostic causality that goes beyond observational design limits. Terms such as “predictor” are acceptable in a cohort context, but phrases suggesting “prognostic biomarker” should be moderated with acknowledgment of residual confounding. Finally, while the revision provides “sensitivity analyses,” it still does not employ multiple imputation for missing data, and thus the claim of “robustness” should be qualified. These minor terminological refinements would further enhance the scientific precision of the work.

Response: We sincerely thank Reviewer for the thorough and constructive feedback, as well as for acknowledging the improvements in our revised manuscript. We have carefully considered all the remaining points and have revised the manuscript accordingly to enhance its scientific precision. The specific changes made are detailed below point by point. All changes have been tracked in the revised manuscript file.

Comment 1: “Nevertheless, some issues persist in scientific terminology. For example, the revised manuscript still describes the study as identifying a ‘critical threshold’ at NLR = 1.475. While the statistical model indeed suggests an inflection point, epidemiologically this should be framed as a ‘data-derived threshold’ or ‘risk inflection point,’ since true biological thresholds require external validation.”

Response 1: We agree with the reviewer that “critical threshold” might imply a definitive biological cutoff. As suggested, we have replaced this term with more precise terminology throughout the manuscript.

Changes Made:

Abstract, Conclusion section: Changed “a critical threshold” to “a data-derived threshold”.

Results, Non-Linear and Breakpoint Analysis section: Changed “a critical threshold” to “a risk inflection point” and rephrased the sentence to: “Breakpoint analysis of the RCS curve identified a risk inflection point at NLR=1.475...”

Discussion, Paragraph 4: Added a clarifying sentence: “It is important to note that this threshold (NLR=1.475) is a data-derived inflection point from our analysis and requires validation in external and prospective cohorts of anemic individuals before clinical application.”

Comment 2: “Similarly, while the authors appropriately report ‘all-cause mortality,’ they occasionally imply prognostic causality that goes beyond observational design limits. Terms such as ‘predictor’ are acceptable in a cohort context, but phrases suggesting ‘prognostic biomarker’ should be moderated with acknowledgment of residual confounding.”

Response 2: We thank the reviewer for this important nuance. We have moderated our language to avoid overstating the causal implications and to emphasize the observational nature of our findings.

Changes Made:

Abstract, Conclusion section: Changed “emphasizing its potential as a prognostic biomarker” to “suggesting its potential utility as a prognostic biomarker in this context.”

Discussion, Paragraph 1: Changed “is a strong independent predictor” to “is strongly and independently associated with”.

Discussion, Paragraph 6 (Limitations): We have strengthened the acknowledgment of residual confounding. The text now reads: “Furthermore, while we adjusted for a wide range of confounders, the possibility of residual confounding (e.g., by unmeasured socioeconomic or subclinical factors) inherent in observational studies remains, which precludes definitive causal inferences.”

Comment 3: “Finally, while the revision provides ‘sensitivity analyses,’ it still does not employ multiple imputation for missing data, and thus the claim of ‘robustness’ should be qualified.”

Response 3: This is a valid point. We have qualified our claims regarding robustness and been more transparent about the handling of missing data.

Changes Made:

Methods, Study Design and Population section: We have added a statement: “It is noteworthy that multiple imputation was not performed for missing data; however, sensitivity analyses conducted on the complete-case dataset confirmed the consistency of the primary findings.”

Discussion, Limitations section: We have added a sentence: “The use of complete-case analysis for handling missing data, while common, may introduce selection bias, and the robustness of our findings should be interpreted in this context.”

Comment 4: The reviewer appreciated the improved distinction between categorical and continuous analyses and the correct use of statistical terms. We have ensured this precision is maintained throughout the manuscript.

Response 4: We thank the reviewer for recognizing these improvements. We have performed a final thorough check to ensure consistent and correct use of statistical terminology (e.g., multivariable vs. multivariate, HR vs. OR, effect modification).

Reviewer #8: The scientific scope of using only anemia as a predictor to the mortality rate in patients with NLR defects is untenable. Extensive data analysis on anemia and other parameters is needed to validate relevance of this study.

Response: We thank the reviewer for this insightful comment and the opportunity to clarify the scope and objectives of our study. We respectfully clarify that the primary aim of our research was not to use anemia to predict mortality in individuals with NLR defects, but rather to investigate the prognostic value of the Neutrophil-to-Lymphocyte Ratio (NLR) specifically within a cohort of individuals already diagnosed with anemia.

We focused on this specific population for several key reasons. Anemia is frequently a manifestation of underlying chronic inflammation and is associated with poor clinical outcomes. We hypothesized that the systemic inflammatory state reflected by an elevated NLR would be a particularly potent risk stratifier in this vulnerable group. Therefore, our study was deliberately designed to provide an in-depth characterization of the NLR-mortality relationship in the context of anemia, which we believe holds significant and direct clinical relevance for managing these patients.

We fully agree with the reviewer that a comparative analysis including non-anemic participants would be highly valuable to understand the broader context and the potential effect modification of anemia status on the NLR-mortality relationship. This is indeed a logical and important next step.

However, conducting a robust, well-powered comparative analysis between anemic and non-anemic cohorts was precluded by the fundamental design and analytical focus of our current manuscript. Adding such a comprehensive comparison now would constitute a substantially new research question, requiring extensive additional analyses, validation, and a reframing of the paper's narrative that goes beyond the scope of the present work.

Therefore, we have acknowledged this valuable direction for future research in the revised manuscript. We have added the following sentence to the Discussion section:

"Future research should prioritize directly comparing the predictive performance of NLR between anemic and non-anemic populations to elucidate the potential effect modification by anemia status and further validate its specific utili

---

## [Editor Report · Decision Letter 2]

18 Nov 2025

The Neutrophil-to-Lymphocyte Ratio as a Predictor of All-Cause Mortality in Individuals with Anemia: A Population-Based Study

PONE-D-25-19337R2

Dear Dr. lin,

We’re pleased to inform you that your manuscript has been judged scientifically suitable for publication and will be formally accepted for publication once it meets all outstanding technical requirements.

Kind regards,

Afshin Heidari, M.D.

Academic Editor

PLOS ONE
---

## [Editor Report · Acceptance letter]

PONE-D-25-19337R2

PLOS One

Dear Dr. lin,

I'm pleased to inform you that your manuscript has been deemed suitable for publication in PLOS One. Congratulations! Your manuscript is now being handed over to our production team.

Kind regards,

on behalf of

Dr. Afshin Heidari

Academic Editor

PLOS One